# Adopting a Holistic Approach to Alcohol Brief Interventions for Women in a Prison Setting in the UK: A Qualitative Exploration

**DOI:** 10.3390/ijerph21121671

**Published:** 2024-12-14

**Authors:** Jennifer Louise Ferguson

**Affiliations:** Law, Policing and Criminology Department, School of Social Sciences, Humanities and Law, Teesside University, Middlesbrough TS1 3BA, UK; jennifer.ferguson@tees.ac.uk; Tel.: +44-01642-4463

**Keywords:** alcohol, women, prison, brief alcohol interventions, ASBI, ABI

## Abstract

Currently, women make up only 5% of the prison population, with 3604 women in prison in the UK compared to 74,981 men. Risky drinking is highly prevalent in both the male and female prison population, however, significantly more females drink in a risky way prior to prison (24% compared to 18% of men). In addition to risky drinking, those entering the criminal justice system (CJS), particularly women, are more likely to suffer from inequalities in society. Such inequalities can be linked to the pains of imprisonment for women. The overall research methods discussed in this paper are qualitative interviews. The interviews were designed after two systematic reviews exploring: the gendered pains of imprisonment and the feasibility and acceptability of women and alcohol brief interventions (ABI) were conducted. Interviews were conducted in an open prison setting, with both female residents and relevant staff and stakeholders. To date there is a dearth of evidence in relation to delivering ABI’s in prison, specifically with women. This research explored the feasibility and acceptability of delivering ABI to women in prison and found that when delivering ABI’s in a prison setting, by underpinning the research with criminological theory, could help women capitalise on the “teachable moment” necessary to induce behaviour change. The findings of the interviews found the intervention was both feasible and acceptable and identified five themes highlighting the women’s journey through prison chronologically to enable a more holistic ABI to be developed in future.

## 1. Introduction

Currently, women make up only 5% of the prison population, with 3604 women in prison in the UK compared to 74,981 men [1]. Risky drinking is highly prevalent in both the male and female prison population, however, significantly more females drink in a risky way prior to prison (24% compared to 18% of men) [2]. In addition to risky drinking, those entering the criminal justice system (CJS), particularly women, are more likely to suffer from inequalities in society [3]. Such inequalities can be linked to the pains of imprisonment for women [4] discussed throughout this article.

Alcohol use disorders (AUD) are health conditions associated with those who drink at a risky level [5]. In the UK, around 26% of adults are said to have an AUD (38% of men and 16% of women) [6]. In addition to the often-discussed health effects, there are a wide range of negative social consequences of alcohol use to consider [7]. One example of such negative social consequences is that risky drinking may lead to many women being involved in crime [8], which may lead to being incarcerated in prison. Therefore, both health and social consequences should be considered in interventions delivered to this population. The intervention for the purposes of this research was an alcohol brief intervention (ABI).

Professionals have been trying to deliver alcohol screening and brief interventions (ASBI’s) since the 1700′s [9]. ABI’s have been being used in practice since the 1980′s [10] and are still being implemented, tailored and evaluated now in a range of settings. Alcohol interventions such as ABIs are carried out in the interests of early detection and secondary prevention of alcohol problems [11]. ABI’s are short, non-confrontational, structured conversations about alcohol consumption that seek to motivate and support individuals to consider their alcohol consumption and plan to reduce their consumption and/or risk of harm [12]. Short interventions such as ABI’s are not intended to promote abstinence, but to reduce an individual’s drinking to within the recommended guidelines [13]. It is hoped then that the aforementioned negative and social consequences will also be reduced [13]. Typically, ABI’s are preceded by screening an individuals alcohol use to identify what level they are drinking at. Such screening enables those who would benefit from being offered brief advice to be identified. The 10 question AUDIT (alcohol use disorders identification test) [14] is considered the most accurate at identifying whether someone is drinking at a harmful level [15,16] and is the gold standard when compared to other shorter tools (AUDIT-C (alcohol use disorders identification test—consumption) [17], CAGE (Cutting down, Annoyance by criticism, Guilty feeling, and Eye-openers) [18], FAST (Fast alcohol screening test) [19]). The AUDIT tool allows a score from 0–40 to be identified by adding up participants answers from the simple ten questions as each question carries a score from 0–4. Once added up the score can then be categorised, with categories ranging from 0 (abstinence), 1–7 (drinking within the recommended limits), 8–15 (harmful drinking), 16–19 (hazardous drinking) and 20 or more (probably dependent) [14]. Both the screening and delivery of the intervention for ASBI’s can be delivered by individuals with no specialist training which means staff in any busy environment can potentially fit this questionnaire into their busy work [20].

As ABI’s aim to motivate individuals to change, capitalising on a “teachable moment” is helpful in aiding change and previous work has shown that presenting in Accident and Emergency departments with an injury and having to wait to be seen is a good example of this [21]. At present, although ASBI’s are well evidenced in health settings such as primary care [22], there is a dearth of evidence in relation to intervention being carried out in the prison setting, and particularly for women in prison. To date there are no interventions tailored specifically for women in an open prison [23]. This could be explained by the small numbers of women in open prisons, yet, in an institution where rehabilitation is necessary, alcohol prevalence is high, and the pains of imprisonment [4,24] are evident, there is a need to explore this further.

There are currently only twelve female prisons in the United Kingdom (UK) and women are only held in one of two conditions; open or closed prison conditions. This means that women are held on average, 64 miles from their home [25], with plenty being held considerably further. Open prisons are more lenient conditions and only house women that have been risk-assessed and are eligible for such conditions. These prisons have minimal security in comparison to closed prisons and allow eligible prisoners to spend time away from the estate on licence (known as return on temporary licence [ROTL]) for activity such as work, education or visiting family. When carrying out research with women in prison, it is important to understand their experiences of incarceration. The pains of imprisonment was first introduced by Sykes [24] who stated that the psychological pain of imprisonment is as damaging to an individual as our more dated forms of punishment which focused on inflicting physical pain and torture. The experience of women in prison can be linked to what Crewe et al. [4] described as gendered pains of imprisonment. Such gendered pains center around: 1. losing contact with loved ones, 2. lack of power, autonomy and control, 3. mental health and physical wellbeing and, 4. issues with trust, privacy and intimacy [4]. Women in prison suffer significant issues compared to their male counterparts and this includes issues such as: more likely to suffer from mental health issues (65% compared to 37% of men) [26], more likely to have experienced emotional, physical or sexual abuse as a child (53% compared to 27%) [26], more likely to have attempted suicide (46% compared to 21%) [27], women have on average two dependent children [27] and are held an average distance of 64 miles from home when in custody [28].

The overall aim of this research was to assess the feasibility and acceptability of carrying out ASBI’s with female prisoners in an open prison in the North East of England. This research combines both public health elements (ABI) and criminological concepts (the pains of imprisonment for women). The advantage of drawing from both perspectives and disciplines provided a richer contextual interrogation of the data and supports the creation of new knowledge. There were research questions set to answer the overarching research aim: what barriers and facilitators are there for alcohol screening and brief interventions (ASBI’s) for women in prison? And, what type of ABI is best for women who are in prison?

## 2. Materials and Methods

This qualitative research was one objective of a larger piece of doctoral research which set out to design a future randomised controlled trial (RCT) of ABI with women in prison. Two systematic reviews (not discussed here) were conducted to help with the design of the interview schedules.

The Medical Research Council (MRC) published a widely used framework in 2000 (later updated in 2008 [29]) with four phases: development, feasibility and piloting, evaluation and implementation [30]. As there is a dearth of evidence surrounding ASBI’s with women in prison, it was necessary to start with stage one of the MRC framework [30] and explore the acceptability and feasibility of ASBI’s with women in prison with the pains of imprisonment at the forefront. This was addressed by carrying out semi-structured interviews with women in prison, as well as staff and stakeholders in a female open prison setting. This qualitative insight allowed the women, staff and stakeholders to share their views on the intervention components and delivery at this early stage of development (Phase 1 of the MRC framework [30].

All face to face interviews were conducted inside the prison. There were specific issues with the nature of this research and the vulnerability of the women involved, therefore both University ethical approval and His Majesty’s Prison and Probation Services (HMPPS) ethical approval needed to be obtained in advance. Online Integrated Research Approval System (IRAS) ethical approval forms had to be signed off by Teesside University ethics committee (Ref: 008/18), before being sent to HMPPS (Ref 2018-364) for further approval, as well as the Governor of the prison approving the research.

The first stage of the qualitative work involved semi-structured interviews with female residents in an open prison (*n* = 12). The prison re-enforced the need for the researcher to refer to the women as ‘residents’. This research used the 10 question AUDIT [14] screening tool as a guide in the conversation. The screening tool was filled in by the women, however the goal was not to assess prevalence and conduct analysis; but get a sense of how feasible and acceptable the women found the questions. Residents were also shown an infographic of the intervention to visualise the intervention and give feedback on the various components of the intervention.

Being a qualitative study, a non-probability sample was used with residents. The research aimed to achieve the maximum variation of perspectives within the sample of female residents. Participants were included if they were residents in the prison and deemed capable of giving consent. It was decided that data saturation for this study would be reached once no substantively new themes emerged from the analysis of three consecutive interviews [31]. Data saturation was deemed to be reached after interviewing 12 residents. The age, nature of offence, length of sentence and history of trauma varied vastly among participants. What did not vary significantly amongst the women was whether or not they had children, and their alcohol use. Out of the 12 women, 11 mentioned that they had children. When retrospectively filling in the AUDIT, all women scored a positive result of 8 or more, and although not being used for screening at this time (simply to aid discussion), the results suggest that all women would have been offered an ABI had this been the next phase of the research.

The next stage of the qualitative work involved semi-structured interviews with relevant staff and stakeholders within the prison. For the purpose of this research, relevant staff and stakeholders were considered to be any individual whose role involves the wellbeing of the women or potential involvement in the roll out of a trial, for example the Governor of the prison. As with the women, staff were shown the intervention and asked questions around feasibility and acceptability. It was also important to get a sense of their experience in relation to alcohol work, and their journey to the role they currently hold.

Both sets of participants were recruited into the study in the same way. The researcher explained the study to the Head of Reducing Reoffending and all staff and women in the prison were informed of being able to take part. Information leaflets were handed out to all staff with a contact number to contact the researcher to arrange an interview. For residents, the women expressed their interest to staff, who then informed the researcher that the women were happy to be approached.

With regards to sampling relevant staff and stakeholders to take part in interviews, a more purposive approach was taken [32]. Participants were included in the study if they worked in the prison, or were relevant stakeholders (responsible for alcohol services within the prison) and were deemed capable of giving consent. It was important to look at gender and role here. The intervention is aimed at females in prison who may have difficulties in trusting a certain gender or authority, therefore it was important to interview staff members of different gender and role within the setting. Semi-structured in-depth qualitative interviews were undertaken with staff and stakeholders. Gender varied across the participants, with three female and three male interviewees. Their roles also varied (Table 1), as did the length of their experience of working in the prison setting.

Before each interview the ability to consent to take part was rechecked and a consent form was filled in, to provide written consent. All interviews then took place in an identified meeting room. Interviews lasted approximately one hour, however some were closer to two hours. All interviews were audio recorded, anonymised and transcribed verbatim. Thematic analysis [33] was chosen to uncover themes arising from the data so as not to impose findings upon the data. This inductive approach was initially chosen as a form of analysis in this research because it was most suited to the aims and objectives, to understand the acceptability and feasibility. All data for this research was stored in accordance with the Data Protection Act 1998 and The General Data Protection Regulation 2016/679.

## 3. Results

A total of eighteen interviews; twelve resident interviews and six with staff and stakeholders were carried out. The Interviews with residents were undertaken between December 2019 and January 2020, prior to COVID-19 pandemic but due to later COVID-19 restrictions within prisons, four of the six staff and stakeholders’ interviews were carried out via telephone.

In the table below are a list of all participants with their ID numbers (Table 1). Although the screening tool was not used to assess prevalence and offer an intervention to participants, the AUDIT scores for the women are also recorded here for reference to give context to their responses. Staff were shown the screening tool but did not fill in the AUDIT in the same way as this would not happen in any future trial, and so AUDIT scores are not included for those participants.

Thematic analysis of the transcripts led to the identification of four main themes that centered around the concept of the women’s “journey” into and out of prison:A woman’s journey into prison;The journey through prison as a woman;Influences on a woman in prisons decision making;A woman’s new journey when she leaves prison;

Using an inductive approach [33], it was interesting to note that both sets of participants views aligned to the same themes. It is acknowledged that this could be because the nature of the questions were very similar; however, what is interesting is that both sets of participants volunteered information and steered the conversation towards issues that were not directly asked about. An example of this is the way in which participants referred to a journey into and through prison, and the importance of the new journey ahead. This chronological “story” is linked to how the women see a shift in their identity due to their sentence. It is interesting because the participants were only asked questions around the feasibility and acceptability of the ABI, who they think should deliver the intervention, and when and where it should be delivered, but woven into their answers, were stories of important key points in the journey through prison that give a rich context to this study and should inform future research.

### 3.1. A Womans Journey into Prison

This theme centered around women discussing their life prior to sentencing. Both the women and staff reflected upon some of the reasons the women had found themselves in prison. The impact of past trauma and the use of alcohol featured predominately here. Many of the women discussed being in “abusive relationships”, either as a victim of childhood trauma or experiences they had had more recently. Some even suggested that such relationships had in fact led to their incarceration.


*“I think most women are in because of a man… they’ve been led astray by a man.” (015)*



*“Everyone’s got shit going on. And d’ya know, I’ve come out the other side” (002)*



*“I could write a book with what I’ve witnessed, it’s been mad, it doesn’t phase me. I deal with it” (016)*


As well as the abuse they suffered, they also noted the impact of their own negative behaviour. Peer pressure or the need to please a partner featured here, with one woman attributing her sentence to a man, but acknowledging that she played a part in the offence because she loved him and would have done anything he asked her to. It is noteworthy to mention that the women expressed that they were not concerned about going to prison; possibly due to the chaotic life that they had in the community. Discussions centered around being in and out of prison for a number of years and sometimes even referring to prison as “a relief”.


*“… you see girls coming in that are coming off the streets… and then you can see them like… maybe even five days later and they’ve had a shower and they’ve washed and they’ve had a bit of food and they look totally different and you think, you know, I feel sorry for them.” (011)*


As ABI’s are known to work best when an individual has what is known as a “teachable moment” [14], it could be assumed that this period of relief for women could be an opportune time to deliver the intervention. However, knowing the women enter with trauma and are fragile according to staff, it is important to note that women are in fact likely to return to this life upon release, and so work done in prison has to address this fact.

Asking women about alcohol is difficult in prison as it is sensitive but also retrospective in nature. Despite this it was essential to explore their alcohol use because there is a high prevalence of risky drinking within this population of women [34]. Fortunately, during the interviews when they were asked about the intervention and screening tools, the women were very open to the questions and were willing to discuss their alcohol use before entering the criminal justice system, with some participants stating that their alcohol use was one of the reasons they had ended up in the prison. The AUDIT was used to facilitate discussion and the researcher found that this caused the women to open up about their drinking in detail, showing screening is a necessary component of the intervention as well as for assessing prevalence. All women scored positive for a potential alcohol use disorder (8+), with a number scoring possible dependence (20+). By carrying out the screening tool with the women, many were surprised with their score and it appeared as though this ten question tool would be an important part of the intervention itself in terms of instilling a small change in the women’s behaviour [35].


*“I didn’t expect it to be as bad as it was (AUDIT score). ‘Cos I didn’t feel like it was as bad as it was.” (005)*


The women suggested that their alcohol use was a “normal” part of their lives prior to prison but for some their views changed throughout prison. This notion of a shift in identity became an important part of the acceptability aspect of the intervention because understanding and getting the timing of the ABI right, can have an impact on their behaviour change.

Links were made between their drinking and their sentence and it was evident that the women often drank in risky ways due to something negative that had happened to them. These events ranged from past abuse by loved ones, to losing custody of their children. Staff were empathetic towards women, showing an understanding of why they may come in to the prison with problems with alcohol and drugs.


*“… and I lost custody of me kids, and then I just had, a drink, you know, because it made me feel better…” (019)*



*“… its to reduce that stigma of people using substances… long term use is because its based on some form of trauma or adverse childhood experience. It’s a perfectly rational coping mechanism for whats going on.” (S006)*


### 3.2. The Journey Through Prison as a Woman

As interviews took place in an open prison, naturally both residents and staff discussed the differences between the open and closed conditions they had both experienced. The findings established open conditions as the most acceptable and feasible environment. Women discussed the transition in many ways and also discussed the various alcohol work they had been offered throughout their journey.

Progressing to the open prison estate is a lengthy process and is something prisoners strive towards due to the less strict conditions and ability to gain more freedom. It would therefore be hypothesized that this would be a positive move for women, however the women in this research highlighted that there are unforeseen challenges with the move to the open estate. The residents discussed how fortunate they felt to be in the open prison now, but noted that it was an extremely difficult transition from the closed prison. This was also apparent from the staff point of view too.


*“You get dragged everywhere in that first week…” (015)*


This highlights the importance of timing when planning to deliver an ABI. If women are overwhelmed with information and kept busy upon arrival, it suggests waiting to deliver the intervention.


*“Still now, still now I could go back to a closed prison and I’d probably find it easier.” (005)*


The most difficult part of the transition appeared to be the new level of trust awarded to the women. Drawing on the pains of imprisonment it is evidenced that women can experience issues with trust [4] and this tends to center around the trust women have in others. An interesting finding here was that the women appeared to have difficulty in the trust placed upon them [4,36,37,38]. Despite the difficulties adjusting, women overwhelmingly talked positively about the open conditions once they had settled.


*“… took some getting used to, getting used to ‘normal life’.” (016)*


This suggests delivering an ABI here would allow women to consider their alcohol use when leaving prison. The open conditions best reflect what the women’s life is going to be like when they leave the prison so it is a good opportunity to discuss their plans for when they have access to alcohol as the behaviour change will mirror more closely their behaviour on the outside.

Women were asked if they had taken part in any alcohol work throughout their prison sentence and what became apparent was this was only undertaken if they were asked to. It was not something the women has sought out themselves. This suggests only those women visually presenting with a need are asked. Consequently, highlighting the missing population of the women slipping through the gaps who are drinking at a harmful level but either did not present as such visually, or alcohol was not directly related to their sentence plan and therefore they were not approached about it.


*“You are asked if you want to, if you say no, nothing else happens.” (001)*


These findings suggest that those women who are drinking at a possibly dependent level are appropriately getting the help that they need and are getting more intense interventions from the drug and alcohol services in the prison. However, those women who would score positive on the AUDIT [14] for a potential AUD would continue through their prison journey without any support for their drinking and perhaps without even knowing that they could benefit from an ABI. Screening every woman in the prison and offering those scoring between 8 and 19 would help to this gap [14,39].

### 3.3. Influences on a Woman in Prisons Decision Making

To explore the acceptability of ASBI, and the fact that the intervention is underpinned by behaviour change and the woman reaching a “teachable moment” [14], the findings suggested that it is important to consider what influences the women to make decisions whilst incarcerated. The findings suggested two factors were important to women making decisions: family influences and staff rapport.

Family influence became a prominent theme when asking women question 10 on the AUDIT, “has a relative or friend or doctor or another health worker been concerned about your drinking or suggested you cut down?” This question prompted all women to discuss their loved ones, in particular children and grandchildren. This reinforces the benefit of using the screening tool, but also as part of the intervention itself to aid discussion and build rapport.

The women discussed wanting to change their drinking habits whilst in prison because they could see upon reflection, the effect their drinking had on their children; something they could not see before their journey into prison. This reinforces the timing of ABI being well suited in an open prison environment in order to achieve the intended behaviour change element of the intervention [40].


*“And he’s like Mum, I’ve gotta say you are a better version of yourself… So even they see all that.” (016)*


It was apparent the women felt comfortable sharing their pain of missing their family members with staff because the staff interviewed reinforced the importance of family to the residents.

The staff rapport with the women was one of the most positive, yet surprising findings of the research. The women stating they would prefer uniformed staff to deliver the intervention is not a feeling shared in similar work with their male counterparts [41,42] and suggests a gendered response to interventions in a prison setting.


*“Its like they [staff] take a different tablet here.” (019)*



*“all of the staff are friendly and we talk to them about sensitive issues anyway… even uniformed” (001)*



*“they’ll have a chat with them, they treat them normal, and us like normal” (016)*


As trust issues are prevalent in the prison estate, it was clearly explained that the women only felt this way in the open prison after building rapport with the staff there. Staff who took part in the qualitative interviews were sometimes surprised by this finding themselves, whilst others commented that this was possibly down to the women finally developing some confidence when reaching the open prison. It is interesting to note that no participants discussed the gender of the staff as being important or affecting their responses. What they discussed centered more about the way the staff treat the women in the open facility, putting trust in them for example. Again, a finding supporting the open prison as an acceptable time to deliver ASBI.

### 3.4. A Woman’s New Journey When She Leaves Prison

In addition to the woman’s journey through the prison system was the concept of the woman’s “new journey”. The discussion of this was centered around women thinking about leaving the prison and looking forward to their lives ahead, outside of prison. Both residents and staff spoke profusely about identity transition throughout their journey through prison. This included looking at the days and months ahead, as well as family and relationships. Discussion also led to thinking about the logistics of this new journey. Leaving prison is not as simple as waving goodbye and closing that chapter in their lives, some women may have never lived alone and therefore need to learn how to support themselves.

With the nature of the open estate forcing the women to look forwards, this appears to have given them space to think about repairing or rebuilding relationships they have with people outside of prison. As these relationships appear to be important to the women, they could be drawn upon in the intervention to bring about change. Thus again reinforcing the idyllic timing of the open prison estate. Alternatively to utilising positive relationships, women discussed the opportunity to they had to reflect upon negative relationships in their lives. Women discussed leaving challenging relationships and feeling very positive about not going back out to those and not including those people in their new journey.


*“… got rid of those demons” (016)*


This finding again can be woven into the intervention work, with women, where necessary, drawing parallels between these negative relationships and their alcohol use.

These qualitative findings suggest women rarely leave the open prison as the same person they were upon entering the criminal justice system. This new identity was not just in relation to their alcohol use but also in a new found strength. Both residents and staff talked about this shift in identity, and the women also commented on how their family had observed this change too. Of importance is that the women needed to have been incarcerated, and for a long time, to be able to have this positive shift in identity. The women stated that this would not have happened had they not been forced into reflecting upon their lives.

An unexpected finding was that some women were in fact grateful for their prison sentence. Whilst acknowledging it had not been an easy journey, the women expressed being grateful for the outcome.


*“So I’ve been grateful for the experience” (016)*


The women appeared grateful because the experience had forced a shift in their identity in a positive way. At the heart of this identity shift was the concept of resilience. Being able to work on personal issues when incarcerated was not just a possibility but was encouraged. This is important to consider why this would be a good setting to deliver ABIas it would fit in to the concept of them “working on themselves”.

The identity transition and new found resilience appeared to be rooted in past trauma. With this in mind, it is fair to predict that many of them would have ended up in worse situations had they not found safety in their imprisonment. This past trauma not only related to alcohol use, but also past relationships. Women described a shift in the way they viewed their lives and attributed this to being due to their sentence, suggesting a teachable moment occurs at this point in their journey.


*“I never thought coming into prison would help me work on meself. In a way, and it’s sad to say it, it’s been a blessing, cos, when you’re in here, there’s nothing you can do but work on yourself… there’s no escaping yourself.” (017)*



*“Yeah, I think I’ve learnt now since being in here, and I know they say, ‘everything’s done for a reason’, and I think, must’ve been a reason why, obviously I committed me crime, but I’ve come to prison, but I think it just makes me look at myself differently, and I think it’s built my confidence up” (011)*


## 4. Discussion

This study used a multidisciplinary approach, exploring the behaviour change element of ABI alongside the pains of imprisonment. Due to ABI being well evidenced in other settings, the research did not seek to test the intervention itself, but instead the focus was to address the acceptability and feasibility of delivering such intervention in an open prison setting.

As the qualitative findings show, a more holistic approach to working with women in prison is needed. Their journey into and through prison is complex and requires attention when working with women. Whilst ABI’s are personalized in nature [20], it appears that with this population, this is necessary, but it is also important to engage with the reality of the women’s experiences into and through prison. Without engaging with the reality of their previous experience, women could look at the intervention from the “safety” of prison, free of distractions, rather than consider the reality of their previous lifestyles. This included women’s journeys to prison, women’s experiences of prison and prison setting (open/closed) and importantly, women’s drinking habits. As understood through the pains of imprisonment [4,24] women might have suffered many different past traumas, have difficulties with trust and enter the prison for different types of crime and for different reasons.

The findings suggested it is both feasible and acceptable, however, it was very clear the findings appear to be only applicable to open prison conditions and could not be implemented in a closed prison environment without further research. An example of this was a surprising outcome that the women identified the prison officers as who they would prefer deliver the intervention. This is something that is different to previous similar research in the closed prison estate with males [41,42]. The excellent rapport between the women and the staff in the open prison was prominent throughout the findings and featured significantly when making recommendations for a future pilot study. This was an important finding as it can be utilised as a facilitator to ASBI. With these findings in particular, it can be argued that prison could provide this “teachable moment” [14] for the women in terms of their identity and resilience, where there is the likelihood that the behavioural change element of the ABI would be possible. However, what does need more thought is the timing of the behaviour change itself. It would appear women have two important teachable moments that both need to occur; one upon finding themselves in prison and dealing with the consequences of their actions, albeit perhaps of a trauma, and the second upon preparing for release from the prison, a more resilient, more confident woman. It would then be this second “teachable moment” that would be capitalised upon and women could be screened and offered an intervention. Both sets of participants agreed that screening for a potential AUD [5] universally in the setting was needed. The timing of the intervention was mostly agreed on too and this was that it should take place either three weeks after arriving in the prison, or prior to leaving the prison; again using that second moment.

## 5. Conclusions

This work has added to the evidence base in both experimental criminology and public health disciplines, demonstrating the benefits of working across multiple disciplines. Importantly the research has demonstrated that ASBI’s are both feasible and acceptable for women in an open prison setting and lays the foundation for future work to understand the use of a public health interventions with women in prison. The research suggests that delivering a public health intervention (ASBI) and underpinning the research with criminological theory (pains of imprisonment) could help women capitalise on the “teachable moment” [14] necessary to induce behaviour change. To date there is no research in the prison setting that tailors any intervention for women based on both public health and criminological theory. The findings of this study could be generalised to other interventions with women in an open prison setting and explored further as there is the potential to use the gendered pains of imprisonment as a background to explore different types of interventions.

## Figures and Tables

**Table 1 ijerph-21-01671-t001:** Participants ID’s from qualitative interviews in prison.

**Participant ID**	**Participant Role**	**Male/Female**	**AUDIT Score**
001	Resident	Female	13
002	Resident	Female	32
004	Resident	Female	35
005	Resident	Female	21
010	Resident	Female	22
011	Resident	Female	12
012	Resident	Female	13
015	Resident	Female	31
016	Resident	Female	9
017	Resident	Female	11
018	Resident	Female	11
019	Resident	Female	40
S001	Prison Officer	Female	Not applicable
S002	Prison Officer	Male	Not applicable
S003	Interim Head of Reducing Reoffending	Female	Not applicable
S004	Activities Manager	Male	Not applicable
S006	Prison Governor	Female	Not applicable
S007	Public Health Commissioner	Male	Not applicable

## Data Availability

The raw data supporting the conclusions of this article will be made available by the authors on request.

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
