# Peer review of "Adopting a Holistic Approach to Alcohol Brief Interventions for Women in a Prison Setting in the UK: A Qualitative Exploration"

_ijerph, 2024, doi:10.3390/ijerph21121671_

Round 1
Reviewer 1 Report
Comments and Suggestions for Authors
The manuscript explores the feasibility and acceptability of alcohol screening and brief interventions (ASBIs) in an open prison for women in the UK. The study uniquely combines a public health approach to ASBIs with criminological concepts, specifically the “gendered pains of imprisonment,” to provide context for these interventions. The qualitative research approach, including interviews with both residents and relevant prison staff, led to the identification of four key themes: the woman's journey into prison, her experience within the prison setting, influences on her decision-making, and the transition into a "new journey" upon leaving prison.
The interdisciplinary approach, bridging criminology and public health, provides a unique and thorough view of the factors influencing female prisoners' attitudes toward ASBIs.
The manuscript effectively structures the thematic findings, which capture both the challenges and positive shifts women experience through the prison journey.
Given the limited research on gender-specific alcohol interventions in prison settings, this work contributes meaningfully to an underserved area, highlighting potential improvements for female prison populations.
Here are some questions for the authors in the hope of helping them improve a very interesting work:
1. Could the authors explain why they chose the ASBI framework over other intervention models?
2. Could the authors elaborate further on the “teachable moments” mentioned? Is there an ideal stage during a prison sentence where ASBIs could have the greatest impact?
3. Given that family influence and staff relationships emerged as major themes, do the authors envision incorporating a structured support system or peer mentoring into the ASBI model?
4. The study includes 12 resident participants and 6 staff/stakeholders, but it lacks specific justification for this sample size. Could the authors introduce a discussion on how data saturation was determined? It could strengthen methodological rigor.
5. The study suggests potential "teachable moments", have the authors considered introducing for example a timeline or flowchart to more clearly delineate when ASBI would ideally be introduced? They could reinforce this by incorporating previous studies on optimal timing for interventions, if of course the length of the text allows.
6. Could the authors further discuss potential challenges in implementing ASBIs in prison? (e.g., varying levels of trauma among participants, trust issues, and logistical constraints, etc.)
7. Could the authors elaborate on how the results might vary in other types of prisons, including locked facilities? Further discussion of this would support the applicability of the results.
8. Could the authors elaborate on how gender-specific pain shapes ASBIs specifically for women?
9. Could the authors further discuss how gender influences trust and receptivity to interventions (e.g., whether gendered staffing influences outcomes)?
10. Could the authors mention previous findings with male inmates to further highlight gender-specific needs?
I thank the authors for this innovative study, which contributes to the reflections on implementing prevention interventions in specific contexts for vulnerable groups.
Author Response
Thank you for your comments. I have used a word document to respond to each comment/point. Please see the attached document.

Reviewer 2 Report
Comments and Suggestions for Authors
Authors has provided detailed work on the Adopting a holistic approach to alcohol brief interventions for women in a prison setting. However, the following comments must be addressed before the manuscript can be accepted for publication.
i. Authors must include the method used for their study, the results obtained and conclusion/recommendation in this order in the abstract section.
ii. Since the pain of imprisonment is one of the major focus of the study, authors should provide detailed information with relevant references on the pains experienced by old prisoners. This will help readers and other researchers understand your work.
iii. Authors provided no information on the statistical analysis performed on information provided by the respondents. This information must be added together with the statistical package used for the analysis of their information.
iv. In the discussion section, authors should endeavour to discuss their results with data obtained from literature in similar studies.
v. Authors must include findings from their statistical analysis in their results and discussion section.
vi. Authors should consider employing the service of an English editor before resubmitting the work. Some words are not correctly written while punctuations are not well used at all.
This study is still in its preliminary form and need to be reworked with full attention.
Comments on the Quality of English LanguageThe manuscript must be reviewed to improve its quality of English language
Author Response
Thank you for your comments. These have been addressed in a separate word document, please see attached.

Reviewer 3 Report
Comments and Suggestions for Authors
The manuscript entitled “Adopting a holistic approach to alcohol brief interventions for women in a prison setting”, intends to demonstrate role of holistic approach to alcohol brief intervention for female prison inmates. On the outset, the theme of the manuscript has immense significance in the context of the female wellbeing, especially the prison inmates, who have typically challenging life experiences. The manuscript is a part of the major work of a doctoral thesis, and this could be a reason as to why certain crucial aspects of the topic has been ignored while drafting the article manuscript. The entire manuscript is aimed at qualitative analysis of female inmates’ wellbeing to ABIs. However, in terms of results there very limited qualitative data to draw any conclusive interpretation. May be the manuscript, being part of the doctoral thesis, is unable to include sufficient data into the article manuscript. The results further, only has few highlighted statements of 5-7 inmates about their emotional wellbeing. Considering the scope and theme of IJERPH, the manuscript does fall into its category, however, the data presented may not qualify for the same. Considering the limited data presented in the manuscript, it is likely to go for a social science journal, unless significant amount of data analysis is demonstrated. Despite our observations, we have raised few major and minor queries as mentioned below, which may help the author to improvise the manuscript quality. In essence, we request the author to kindly consider rewriting the entire manuscript, taking into consideration, the following points.
1. ithenticate plagiarism report suggests 65% plagiarism, which is beyond acceptable limits. However, it major (53%) of it comes from the author’s thesis. We request to the author to rewrite the manuscript, rather than extracting lines from the thesis. The current method has resulted in poor outcome of the draft manuscript. Writing afresh as an independent manuscript may give better perspective of clearly defined introduction, objectives, methodology, and results.
2. In the introduction, there is no clear explanation about what is ABI. What is the significance of ABI, and various methods of ABI?
3. Kindly add a flow chart including the inclusion and exclusion criteria, for better visibility of the methodology.
4. Kindly provide the details of the questionnaire as supplementary data.
5. The qualitative analysis in the form of the questionnaire also has certain criteria for analysis, may be in terms of grading systems. The methodology lacks that clarity. Likewise, the expected interpretation of the grading system also should be mentioned in the methodology, or may be in the results. The methodology must include detailing of MRC framework. For what does AUDIT stand? Kindly add the detailing of AUDIT as well.
6. There is significant disparity between various population groups across ethnic and regional lines. Hence, it is important to highlight the ethnicity of the population in the title of the manuscript. If the prison inmates belong to multiple ethnicities, then it is important to take that into consideration.
7. The content of the methodology has several repetitions, and the content is focused on general setup of the methodology, rather than highlighting the actual inclusion and exclusion criteria, and the questionnaires and grading systems.
8. Line 82, a research article is expected to be an independent publication, kindly refrain from mentioning that the study is part of the doctoral thesis.
9. There is significant difference between the inmates and the staff, since the staff have families and freedom of the outside world. How can both be generalized, and compared or ABIS?
10. There is no explanation as to why AUDIT score is not applicable to staff (table 1)? If it is not applicable to staff, then what is the reason for their addition in the table 1?
11. Based on the context of the manuscript, it would be apt to combine the results and discussion into a single section. In the current manuscript, the interpretations are skewed into results section, whereas, the discussion section does not elaborate on the possible interpretations of the results.
12. Full forms of several terms have not been mentioned in the manuscript, e.g., RCT (line 83), MRC (line 87) etc.,
13. Line 97, kindly correct the typo error.
Comments on the Quality of English LanguageMinor grammatical and punctuation errors have been identified throughout the manuscript.
Author Response

(The authors gave the same response as above.)

Reviewer 4 Report
Comments and Suggestions for Authors
I thank the authors for their work and for submitting this paper that contributes to the extension of the current literature. The methodology is interesting and rigorously performed. The hypotheses are clearly expressed and the analyses are adequate for the intended purposes. Last but not least, the results contribute in an interesting way to the literature in this specific area, also providing interesting ideas for practice and research.
I ask the authors if it is possible to better delineate the differences between males and females regarding alcoholism problems, verifying the greater risk in the female gender both in the general and clinical population and in the prison population. Furthermore, I would like to better understand which psychological factors could make a woman, and a woman who ends up in the judicial circuit, more at risk of alcoholism.
Can you better define what "open prison" means?
I believe that the authors could better describe why it is important to intervene in subjects with alcoholism in the prison context and describe some important associated psychological characteristics that could be targets of psychotherapeutic treatment or be a facilitator/obstacle of success, such as alexithymia and emotional regulation (https://doi.org/10.3389/fpsyg.2024.1356024).
Author Response

(The authors gave the same response as above.)

Round 2
Reviewer 2 Report
Comments and Suggestions for Authors
No comment
Author Response
I can see no further comments. Thank you.
Reviewer 3 Report
Comments and Suggestions for Authors
1. The full forms of abbreviations were missing in the previous manuscript which has been rectified in the revised manuscript. However, it is wrong to put the full forms within the brackets instead of the abbreviations. Kindly rectify it.
2. Overall, grammar errors still exist, for instance, line no. 53-54, 58-60, 68-71, 93-99, of revised manuscript.
3. There is a lack of clarity in the usage of the term ASBI and ABI. Kindly address it.
4. The English language needs significant upgradation. Many a times, the sentences are not being meaningful. As previously suggested, we strongly suggest the author to improvise the manuscript in terms of communication.
5. Regarding query no. 6, it is extremely important to mention the ethnicity of the subjects under study, in addition to their jail terms, age, history of alcoholism. Humans of different ethnic groups behave and respond differently in various circumstances. So, for any research manuscript addressing human beings, it is important to highlight the ethnicity of the individuals.
6. Regarding the methodology, there is still no clarity, query no. 5 and 7 remains unaddressed.
7. Regarding query no. 8, manuscript writing for journals has certain guidelines keeping the readers in mind. Good number of the research articles probably are part of PhD theses, but they are not mentioned in the journal manuscript.
8. Essentially, the research article manuscript is a qualitative analysis of ABI for female inmates in UK prison setup, however, the title lacks clarity on that aspect. We suggest that the authors modify the title of the manuscript, clearly indicating that it is a qualitative analysis for female inmates in UK.
9. Ultimately, there is no mention of the detailed questionnaire as per AUDIT, neither in the methodology or in supplementary data, as was raised in query no. 5. Since there is no other data based on which a conclusion can be drawn, it is important to mention the details of the questionnaire.
Comments on the Quality of English LanguageThe revised manuscript has not been modified significantly in terms of English language grammar and punctuations, except that which we pointed out. The manuscript requires significant modification in terms of English language
Author Response
Thank you for your comments, please see the attached word file.

Reviewer 4 Report
Comments and Suggestions for Authors.
Author Response
I can see there are no comments to address. Thank you.